# FFT-Based Angular Compression for CSI Feedback in Single-User Massive MIMO Systems

**DOI:** 10.3390/s25154544

**Published:** 2025-07-22

**Authors:** Felipe Vico, Helen Urgelles, Jose F. Monserrat, Yiqun Ge

**Affiliations:** 1Universitat Politècnica de València— iTEAM, 46022 Valencia, Spain; fevibon@iteam.upv.es (F.V.); heurpe@doctor.upv.es (H.U.); 2Huawei Technologies Company Ltd., Shenzhen 518129, China; yiqun.ge@huawei.com

**Keywords:** 6G, T-MIMO, FFT, CSI feedback

## Abstract

Massive MIMO has emerged as a key enabler in modern wireless communication, delivering unparalleled spectral efficiency and connectivity. Yet, as antenna arrays become larger, significant obstacles arise in handling channel state information (CSI) feedback and the computational burden. This paper proposes a groundbreaking angular-domain transmission method that transitions from the conventional time–frequency domain to the angular domain. By employing projection-based transforms, akin to the FFT-based OFDMA model that introduced frequency-domain transmission with subcarriers, this technique exploits the inherent sparsity of massive MIMO channels in the angular domain, enabling data flows to be seamlessly mapped onto physical paths or rays. The resulting sparsity reduces signaling overhead and streamlines system complexity, making massive MIMO viable for next-generation networks. Simulation and empirical studies highlight how angular-domain strategies reduce feedback requirements, support Tera-bps data rates, and facilitate scalable designs for ultra-large-scale MIMO.

## 1. Introduction

Large-scale or ultra-large-scale Multiple-Input Multiple-Output (MIMO) systems have become a key innovation in modern wireless communication. By significantly increasing the number of Base Station (BS) antenna elements, these systems allow for highly directional signal transmission, similar to focusing a beam of light. This leads to notable improvements in signal quality and reliability for User Equipment (UE). When a BS has many more antennas than a UE, it can better suppress interference and enhance spectral efficiency, resulting in higher data rates and more stable connections for users.

One important example of this technology is Tera-bps MIMO (T-MIMO), which commonly operates in the 10–14 GHz frequency range or even higher in the millimeter Wave (mmW) bands. Each frequency range offers its own benefits and trade-offs. For example, the 10–14 GHz band supports wider channel bandwidths than lower frequencies, enabling higher data throughput while offering moderate coverage. In contrast, mmWave frequencies can deliver extremely high data rates but are more limited in coverage and vulnerable to signal blockage.

However, deploying large antenna arrays across these wide bandwidths also brings challenges, especially regarding energy use, system complexity, and sustainability. Large-scale BSs require substantial power, emphasizing the importance of developing energy-efficient hardware and intelligent protocols. To address these challenges, researchers and engineers are working on advanced hardware designs and dynamic network management strategies to adjust resource allocation based on real-time user needs.

In addition, control signaling, used to coordinate communication between BS and UE, also increases with more antennas and broader bandwidths. If not carefully managed, this signaling overhead can reduce the performance gains from higher spectral efficiency. Therefore, advanced signal processing techniques and optimized BS-UE coordination methods are essential to fully realize the benefits of large-scale MIMO while keeping the system efficient and sustainable.

To address the feedback overhead challenge, we propose a compression strategy in the angular domain for Channel State Information (CSI). By transforming the MIMO channel matrix using a two-dimensional Discrete Fourier Transform (DFT), its sparse structure becomes apparent, allowing substantial compression without losing critical channel information. This allows effective beamforming at the BS while significantly reducing feedback and computational load. Overall, this approach enhances the practicality and performance of future wireless systems.

**Problem Statement.** Although a variety of angular-domain CSI compression schemes have been proposed, ranging from compressive-sensing optimizers that require multiple pilot rounds and iterative solvers (e.g., Ma et al. [1]), to deep-network feedback methods that incur substantial on-device inference cost and large codebook storage at the UE (e.g., Hu et al. [2], Kim et al. [3])—they suffer from one or more of the following drawbacks:High UE complexity. Iterative CS algorithms and unfolded networks demand multiple FFT–IFFT operations, matrix inversions, or gradient steps per frame, exceeding low-power UE capabilities.Training and codebook overhead. Learned compression schemes require extensive offline data, non-trivial model updates, and storage of large neural networks or quantized codebooks at the UE.Limited adaptability. Fixed network architectures or codebooks generalize poorly to new propagation conditions (e.g., dynamic richly scattered NLoS channels) unless retrained or manually retuned.

Our solution is a purely linear training-free FFT-based compression strategy that

Transforms the MIMO channel into the angular domain via a single 2D FFT,Selects angular beams to retain a fixed fraction of the total energy (no loops or learned thresholds),Precomputes end-to-end precoders/decomposers to eliminate any per-frame SVD or iterative solver.

This approach reduces UE complexity to O(NlogN) per FFT, requires no codebooks or training data, and automatically adapts to instantaneous channel sparsity—making it especially well suited to real-time resource-constrained single-user massive MIMO deployments.

The remainder of this paper is organized as follows. Section 2 presents a review of the relevant prior work. Section 3 offers background on MIMO systems and associated decomposition techniques. Section 4 introduces the concept of MIMO sparsity, which forms the basis of the proposed approach. Section 5 and Section 6 elaborate on the Fast Fourier Transform (FFT)-based compression strategy, first from a conceptual standpoint and subsequently through block diagram representations and a detailed protocol description. Section 7 reports the performance evaluation conducted in indoor and outdoor scenarios. Finally, Section 9 summarizes the main contributions and concludes the paper.

## 2. Related Works

A range of approaches has been developed to simplify the acquisition of CSI and to refine channel estimation. In MIMO networks, for example, one common strategy involves using DFT-based feedback, where the receiver relays information about the state channel back to the transmitter through DFT-derived techniques. This feedback underpins essential adjustments, such as beamforming and precoding, that optimize communication performance.

In [4], a practical comparison is made between an explicit CSI feedback scheme (time-domain compression and Principal Component Analysis (PCA)) and the standard NR Rel. 15 Type II CSI. By employing a grid-of-beams precoder to compress the channel dimension and exploit sparsity, time-domain compression offers around a 14% performance gain for low-rank transmissions, albeit at higher UE complexity.

Targeting the high computational load of Zero Forcing (ZF) precoding in large-scale massive MIMO (mMIMO) Orthogonal Frequency Division Multiplexing (OFDM) systems, the authors of [5] propose efficient techniques that approximate time-domain ZF using the block-Toeplitz structure of the channel matrix. Their schemes include an FFT-based conjugate gradient method and a block-Toeplitz QR decomposition, extending naturally to Regularized Zero-Forcing (RZF).

In [6], Beam Space Channel Estimation (BSCE) leverages multiple DFT matrices to suppress non-dominant beams, achieving a 53% cell throughput improvement under 0 dB Signal-to-Noise Ratio (SNR). Unlike dimension-reduction methods, this approach focuses on refining the channel estimation accuracy. Similarly, the authors of [7] introduce BSCE for mMIMO mmW systems (BEACHES), relying on adaptive denoising (SURE) to match state-of-the-art performance in both Line of Sight (LoS) and Non-Line of Sight (NLoS) channels and featuring a hardware-friendly Field-Programmable Gate Array (FPGA) implementation.

Addressing beam selection under limited Radio Frequency (RF) chains, the authors of [8] adopt a support detection strategy that breaks down beamspace channel estimation into small sub-tasks. Each detected sparse channel component is removed from the global estimation problem, reducing pilot overhead and preserving accuracy at low SNR. The authors of [9] tackle beam squint in wideband mmW mMIMO systems via a compressed sensing approach that exploits the frequency-dependent sparse structure. Employing an orthogonal matching pursuit algorithm to capture path power and support reduces pilot overhead and significantly improves the estimation accuracy.

The paper [10] addresses the significant computational and memory challenges associated with uplink detection in massive MIMO systems. To mitigate these issues, the authors propose a novel detection framework that operates in the angular domain, leveraging the inherent sparsity of massive MIMO channels. By applying an FFT, the CSI is compressed by selecting dominant angular beams, thereby substantially reducing the data volume required for processing. The proposed method integrates a linear detection scheme followed by a low-complexity non-linear post-processing step to enhance performance. This approach offers a flexible trade-off between detection accuracy, memory usage, and computational cost. Evaluations using real-world measured channels demonstrate that the method outperforms traditional antenna-domain techniques like ZF, achieving up to 73% reduction in computational complexity and 75% savings in memory without sacrificing, and in many cases improving, detection performance. Furthermore, the authors present a VLSI implementation in 28 nm CMOS technology, which confirms the practicality of their design in achieving high throughput and energy efficiency in real hardware.

Finally, the authors of [11] propose an efficient hybrid precoding and channel tracking strategy for mmW massive MIMO systems, addressing the challenges of hardware cost and channel estimation complexity due to the high number of antennas. The authors leverage the sparse nature of mmW channels in the angular domain to reduce the required number of RF chains through a technique called spatial rotation, which focuses signal power on fewer spatial beams. To minimize interference, a novel scheduling method called Angle Division Multiple Access (ADMA) groups users based on their direction of arrival (DOA). For channel tracking, the paper introduces a technique that separates channel state information into DOA and gain components; it uses a modified unscented Kalman filter (MUKF) for DOA tracking and beam training for gain estimation. This approach significantly lowers training overhead and computational burden while maintaining high tracking accuracy. Simulation results demonstrate that the proposed method approaches the performance of full digital systems and outperforms existing beam selection and channel estimation methods, particularly in terms of achievable sum rate and tracking precision, even under mobility and practical limitations.

### Recent Advances in Angular-Domain Processing and CSI Compression

In the past few years, a consensus has emerged that exploiting the inherent sparsity of massive MIMO channels in the angular domain—via an explicit 2D FFT—can dramatically reduce CSI feedback overhead without sacrificing estimation accuracy. Ma et al. embed a learned linear projection into a compressive-sensing optimizer, achieving near state-of-the-art CSI reconstruction with minimal computational cost [1]. Building on this idea, Hu et al. unfold the iterative shrinkage-thresholding algorithm into a trainable network (LORA), learning both regularization weights and shrinkage thresholds to gracefully adapt to varying feedback rates [2]. Beyond model-driven designs, Kim et al. recast CSI feedback as a lossy source-coding problem with side information, using a diffusion process at the BS to reconstruct the channel from a compact codebook at the UE—halving the required feedback rate compared to prior deep-network methods [3].

Robustness to deployment variation has also been addressed: Wang et al. introduced EG-CsiNet, which explicitly decouples and aligns each path’s parameters (angle, delay, gain) prior to encoding, yielding over 3.5 dB improved CSI fidelity when tested in previously unseen environments [12]. A broader tutorial on extremely large-scale MIMO by Wang et al. demonstrates that, as array apertures grow, the channel’s Fourier plane–wave spectrum becomes increasingly sparse—providing a rigorous physical justification for 2D FFT-based compression [13]. Complementing these more recent works, the survey by Wang et al. reviews fundamentals, challenges, and solution frameworks for XL-MIMO—covering near-field channel modeling, performance analysis, and hardware architectures—which underpins and motivates angular-domain CSI compression strategies in practice [14].

Together, these advances confirm that FFT-based beam-domain transforms—augmented by lightweight learning or generative schemes—offer a practical and scalable path to high-fidelity CSI feedback in next-generation massive MIMO systems.

## 3. Massive MIMO System and Decomposition

Massive MIMO systems were introduced in 3GPP Release 15 and refined in Release 17 and subsequent releases. T-MIMO is poised to be a foundational component of forthcoming Sixth Generation (6G) technologies, leveraging an extensive array of antenna ports at the BS to enhance spectral efficiency significantly. In this context, the term large-scale or ultra-large-scale MIMO denotes wireless systems with a high count of BS antenna ports, thereby improving spatial diversity and multiplexing while posing substantial challenges to CSI feedback mechanisms.

Under the traditional MIMO + OFDM paradigm, the MIMO channel for a subcarrier is represented by a frequency-domain channel matrix H. As the number of antenna ports increases, the matrix grows accordingly, making direct transmission of all its elements infeasible due to bandwidth limitations. Conventional MIMO precoding methods rely on the UE to estimate H (comprising 2N by 2M ports, with N ports per polarization at the BS and M per polarization at the UE). This full matrix is then compressed, either implicitly via parameters such as Reference Signal Received Power (RSRP), Precoding Matrix Indicator (PMI), Rank Indicator (RI), and Channel Quality Indicator (CQI) (3GPP Release 15) or explicitly in the matrix product format introduced in Release 16, before being sent back to the BS. Upon reception, the BS decompresses H to restore its original dimensions before generating the precoding matrix. This procedure incurs substantial overhead owing to the size and redundancy of **H**, compounded by the compression–decompression cycle, necessitating a careful balance between CSI accuracy and feedback costs.

Accurate feedback of H is pivotal for advanced precoding techniques that refine signal transmission based on real-time channel characteristics. However, the process by which the UE measures CSI Reference Signals (CSI-RS) over the downlink and transmits the corresponding channel matrix H through the uplink consumes considerable bandwidth.

Consequently, new approaches to CSI feedback and precoding are required to handle these high-dimensional channel matrices while ensuring scalability and efficiency. By innovating beyond traditional compression and decompression schemes, it is possible to reduce overhead and computational demands, thereby advancing the performance and feasibility of large-scale or ultra-large-scale MIMO in next-generation wireless systems.

### 3.1. MIMO Decomposition

MIMO channel technology uses precoding and decoding matrices to transform a multi-antenna (MIMO) channel into multiple parallel Single Input Single Output (SISO) sub-channels (or MIMO streams). In centimeter wave (cmW) bands, not only strong LoS rays but also significant NLoS paths can be leveraged for transmission. Unlike mmW systems that mainly rely on analog beamforming for LoS paths, cmW MIMO uses digital precoding at the transmitter and decoding at the receiver. By applying these matrices, the original MIMO channel is decomposed into several orthogonal SISO sub-channels of different SNRs, achieving spatial multiplexing and boosting both efficiency and reliability.

Singular Value Decomposition (SVD) is a key mathematical tool for this decomposition. For an M × N channel matrix H (with M transmitter antennas and N receiver antennas), SVD factors H into(1)$$H=U\Sigma V^{H},$$
where U and V are unitary matrices, and Σ is a diagonal matrix of singular values. Precoding by V and decoding by $U^{H}$ transform the transmission into parallel SISO sub-channels, with each diagonal element in Σ indicating the channel quality (SNR) of one sub-channel.

The number of non-zero singular values, called the MIMO rank (r), is at most the smaller of M and N, but in practice it is usually much lower, limited by propagation environments that reduce the number of effective paths. Increasing the number of antenna ports at both ends can lift this theoretical rank limit and improve singular values, thereby increasing overall spectral efficiency. This principle underpins the evolution of MIMO systems with ever more antennas in successive wireless generations.

### 3.2. Problems in Large-Scale MIMO Channels

MIMO decomposition requires significant hardware resources and power, partly because adding more antenna ports increases the volume of CSI feedback. In Fifth Generation (5G), for example, the BS transmits CSI-RS signals (pilots) to the UE; the UE estimates the channel and encodes this into CSI, which it sends back over the uplink. The BS then decodes this CSI to determine a suitable precoding matrix for the subsequent downlink transmission. This process relies on accurate CSI measurement, encoding, feedback, and decoding. However, maintaining such accuracy in time-varying wireless environments is challenging and expensive regarding radio resources, especially as the number of antenna ports grows.

From Fourth Generation (4G) to 5G to 6G, the number of antenna ports has steadily increased to boost MIMO spectrum efficiency. While UE antenna ports might rise from 8 to 32 or even hundreds, the BS side can be far more aggressive, reaching up to 1024 antenna ports in 6G T-MIMO systems. Moreover, T-MIMO aims to use much wider bandwidths (e.g., 200–400 MHz), which further complicates CSI measurement because the channel matrix can vary significantly across subcarriers. If T-MIMO simply replicated 5G’s CSI feedback protocols, a large portion of the uplink would be consumed by CSI reporting, creating a significant bottleneck.

Addressing these challenges calls for innovative CSI feedback and precoding strategies that can handle high-dimensional channel matrices while maintaining scalability and efficiency. We propose capitalizing on the inherent sparsity of massive MIMO channels by concentrating on their angular-domain components. This method reduces feedback overhead without sacrificing CSI accuracy and streamlines SVD computations, alleviating the computational burden on the base station. Consequently, it introduces a more efficient paradigm for encoding and decoding CSI feedback in massive MIMO systems, thereby driving advancements in wireless communication technologies.

### 3.3. Proposed FFT-Based Compression: Novelty and Advantages

Whereas recent angular-domain CSI compression schemes rely on learned projections [1], unrolled iterative networks [2], diffusion-based reconstructions [3], or explicit path-parameter alignment at the UE [12], our method is entirely training-free and linear. We exploit a plain 2D FFT followed by simple energy-thresholding to identify the small set of active angular directions, thereby exposing the channel’s low-rank structure without any neural-network inference or codebook design at the UE.

Building on this sparse FFT-domain support, we derive an approximate SVD of the full channel by computing the dominant singular vectors only on the masked submatrix. Crucially, because both the FFT and the subsequent truncation and SVD steps are unitary or linear, we can precompute end-to-end transmit and receive matrices that fold FFT, mask, and truncated SVD into fixed precoders/decomposers—eliminating any per-frame matrix factorizations or iterative solvers.

Finally, we detail a three-phase protocol (omnidirectional pilot + FFT support detection → directional pilot on selected beams → data transmission) that replaces costly beam sweeps with just a handful of omni pilots and FFTs. We validate this pipeline over realistic ray-traced channels at 12–30 GHz, showing near Shannon-limit capacity, while requiring only a 0.5–5 μs 2D FFT per subcarrier—an overhead easily amortized over tens of milliseconds of coherence time.

By eschewing training, deep models, or large codebooks, our FFT-only approach achieves high-fidelity CSI feedback and low-complexity precoding in next-generation single-user massive MIMO systems, complementing the physical insights on XL-MIMO sparsity recently surveyed in [13,14].

## 4. Sparsity and Physical Modeling in MIMO Transmission

Ultra-large-scale MIMO refers to wireless systems employing hundreds or even thousands of antennas, typically operating at cmW or mmW frequencies. These high frequencies allow for compact antenna spacing, making it possible to gather fine-grained spatial information about signal propagation. Despite the massive number of antenna elements, signal propagation remains dominated by a limited number of strong paths or “clusters.” These dominant directions arise from interactions with key environmental features and lead to a sparse and low-rank representation of the channel matrix in the angular domain.

A rigorous and physically grounded method for characterizing this phenomenon is ray-tracing. This approach models the channel as a summation over discrete propagation paths (or “rays”) that may involve various environmental interactions, such as reflection, refraction, and scattering. Each ray represents a potential eigenchannel of the system, corresponding to a particular direction, delay, and attenuation level.

### 4.1. Physical Basis of Sparsity via Ray-Based Modeling

Ray-tracing techniques offer a powerful framework for capturing the complexity of electromagnetic wave propagation in real-world environments. Each significant propagation path can be associated with a dominant eigenchannel in the system’s singular value decomposition (SVD), and these eigenchannels can be classified as

Principal eigenchannel: typically represents the line-of-sight (LoS) path.Secondary eigenchannels: represent non-line-of-sight (NLoS) paths with single reflections.Higher-order eigenchannels: involve multiple reflections and exhibit greater delay and attenuation.

Although the total number of rays increases with the environmental complexity, their signal contributions decay with each additional interaction. Thus, only a few rays, often aligned with physical structures, meaningfully affect the received signal, reinforcing the sparse and low-rank nature of the channel matrix. Figure 1 depicts the relation between physical routes and the propagation rays.

### 4.2. Channel Matrix Representation and Eigenchannel Decomposition

The channel matrix H can be understood to analyze these propagation phenomena as the result of multiple dominant rays, each corresponding to a physical path. These dominant rays define the primary eigenchannels of the matrix H, which can be effectively decomposed using the SVD (Equation 1).

The dominant eigenchannels in this decomposition correspond to the most significant propagation paths. In practice, only a few such eigenchannels carry most of the signal energy, and the corresponding singular vectors U and V represent the optimal receive and transmit directions, respectively.

### 4.3. Implications for Signal Processing

The spatial sparsity inherent to ultra-large-scale MIMO systems significantly simplifies signal processing tasks. Since energy is concentrated in just a few angular directions, the channel matrix exhibits compressibility, making it amenable to efficient approximations using, for example, the FFT. This structural property facilitates scalable channel estimation, beamforming, and feedback mechanisms crucial for next-generation wireless systems.

## 5. FFT-Based Matrix Compression

### 5.1. Overview of the FFT Compression Technique

Electromagnetic propagation can be modeled with high accuracy using ray-tracing techniques at carrier frequencies exceeding 10 GHz, corresponding to wavelengths shorter than 3 cm. In this regime, dominant scatterers and reflectors in the environment are typically much larger than the wavelength, and the effects of diffraction and interference become secondary. This physical insight underpins spatial transformations to exploit angular sparsity in the propagation channel.

We consider a massive MIMO (mMIMO) system characterized by the canonical input–output model:(2)y=Hx+w,
where $x\in \Phi^{M}\subset C^{M}$ denotes the transmitted signal vector, $y\in C^{N}$ is the received signal, w represents additive white Gaussian noise, and $H\in C^{N\;\times \;M}$ is the channel matrix. This model applies per subcarrier in an OFDM framework for wideband channels, where the available bandwidth is partitioned into orthogonal narrowband subchannels.

Assuming rectangular array structures with dimensions n × m at the transmitter and $n^{\prime }\;\times \;m^{\prime }$ at the receiver, we express M=nm and $N=n^{\prime }m^{\prime }$ and reshape the transmit and receive vectors into matrices:(3)$$x\in C^{n\;\times \;m},\;y\in C^{n^{\prime }\;\times \;m^{\prime }}.$$

We denote individual entries of these matrices as $x_{i,j}$, $y_{i,j}$, and $w_{i,j}$, where i=1,…,n (or $n^{\prime }$), and j=1,…,m (or $m^{\prime }$).

To facilitate angular-domain analysis, we introduce the 2D discrete Fourier transform (2D DFT) operator $F_{2}$, which maps complex-valued matrices of size n × m to their frequency-domain counterparts:$$F_{2}:C^{n\;\times \;m}\to C^{n\;\times \;m}.$$

The 2D DFT is explicitly given by(4)$${\hat{X}}_{k,\ell }=\frac{1}{\sqrt{nm}}\sum_{i=0}^{n-1}\sum_{j=0}^{m-1}x_{i,j}\cdot e^{-2\pi i\left(\frac{ki}{n}+\frac{\ell j}{m}\right)}.$$

**Remark** **1**(Fast Fourier Transform (FFT))**.**
* The 2D DFT can be efficiently implemented using the FFT algorithm, with computational complexity O(NlogN), where N=nm. This enables low-complexity implementation of both precoding and detection processes.*

**Definition** **1**(Sparse and Highly Sparse Operators)**.**
*We define a matrix $H_{1}\in C^{N\;\times \;N}$ as sparse if the number of non-zero elements is N·p with p≪1. A matrix $H_{2}$ is termed highly sparse if most of its rows and columns are entirely zero, implying that non-zero entries are confined to a small localized submatrix.*

### 5.2. Sparsity of the Equivalent Channel via 2D FFT

At high frequencies where ray-tracing assumptions hold, the physical propagation channel exhibits inherent angular sparsity. That is, only a limited number of angular directions contribute significantly to signal transmission and reception. This property manifests in the transformed domain when the channel is pre- and post-processed using 2D DFTs:(5)$$\tilde{H}=F_{2}^{-1}HF_{2}.$$

Here, $F_{2}\;\in \;C^{nm\;\times \;nm}$ denotes the unitary 2D DFT operator acting on vectorized n × m matrices, and $F_{2}^{-1}={\left(F_{2}\right)}^{H}$ is its inverse (the 2D IDFT).

**Remark** **2****.**
*Under angular sparsity assumptions, the transformed matrix $\tilde{H}$ is highly sparse, containing non-negligible values only in a small subset of rows and columns. This reflects the physical constraint that only a few spatial directions carry significant energy.*


### 5.3. Low-Complexity MIMO Detection in the Sparse Domain

The impact of this transformation on the MIMO detection problem. Applying the 2D DFT to the transmitted signal and its inverse to the received signal yields(6)$$F_{2}^{-1}y=F_{2}^{-1}\left(HF_{2}F_{2}^{-1}x+w\right)=F_{2}^{-1}HF_{2}\tilde{x}+\tilde{w}.$$

Since the DFT is a unitary transformation, $\tilde{w}=F_{2}^{-1}w$ remains a white Gaussian noise process. We define$$\tilde{H}=F_{2}^{-1}HF_{2},\;\tilde{y}=F_{2}^{-1}y,\;\tilde{x}=F_{2}^{-1}x,$$
leading to the equivalent system:(7)$$\tilde{y}=\tilde{H}\tilde{x}+\tilde{w}.$$

Given that the DFT is unitary, $\tilde{w}$ retains the statistical properties of white Gaussian noise. The key advantage of this transformation is that detection algorithms can now operate on the significantly smaller support of $\tilde{H}$, reducing complexity while preserving accuracy.

For example, zero-forcing detection in the angular domain yields(8)$${\tilde{x}}_{0}={\tilde{H}}^{+}\tilde{y},\;\mathrm{where}\;{\tilde{H}}^{+}={\left({\tilde{H}}^{H}\tilde{H}\right)}^{-1}{\tilde{H}}^{H}.$$

#### 5.3.1. Step 1: Detection of Active Angular Directions

To exploit the sparsity of $\tilde{H}$, it is first necessary to identify the indices corresponding to its non-zero rows and columns, which represent the active receive and transmit angles, respectively.

##### Step A: Receive Angle Detection

An omnidirectional signal is transmitted, and the receiver applies a 2D DFT to the received signal matrix. The magnitudes of the resulting coefficients are inspected, and those exceeding a threshold ϵ are deemed active.$$\mathrm{If}\;|{\tilde{y}}_{i}|\;>\;\epsilon ,\;\mathrm{then}\;\mathrm{direction}\;i\;\mathrm{is}\;\mathrm{active}.$$

This process effectively identifies the non-zero rows of $\tilde{H}$.

##### Step B: Transmit Angle Detection

A reciprocal process is performed for transmit angle identification. Leveraging channel reciprocity, the roles of transmitter and receiver are reversed, and the active directions correspond to the non-zero columns of $\tilde{H}$, or equivalently, the non-zero rows of ${\tilde{H}}^{T}$.

The matrix–vector multiplication below illustrates how the sparse structure of $\tilde{H}$ leads to output sparsity when the input signal x has uniform non-zero entries:(9)$$\tilde{H}x=\left[\begin{array}{cccccccccc} • & \; & & \; & & \; & & \; & & \\ \; & \; & & \; & & \; & & \; & & \\ \; & \; & & \; & & \; & & \; & & \\ \; & • & \; & & \; & • & \; & & \; & \\ \; & \; & & \; & & \; & & \; & & \\ \; & \; & & \; & & • & \; & & \; & \\ \; & \; & & \; & & \; & & \; & & \\ \; & \; & & \; & & \; & & \; & & \\ \; & \; & & \; & & \; & & \; & • & \\ \; & \; & & \; & & \; & & \; & • & • \end{array}\right]\left[\begin{array}{c} •\\ •\\ •\\ •\\ •\\ •\\ •\\ •\\ •\\ • \end{array}\right]=\left[\begin{array}{c} •\\ •\\ •\\ •\\ • \end{array}\right].$$

This two-step angular-domain filtering forms the basis for reducing the channel matrix to a low-rank approximation, enabling efficient detection algorithms in subsequent stages.

### 5.3.2. Step 2. Standard MIMO Detection on the Reduced Subspace

After Step 1, only a reduction in the dimensionality and the application of a standard detection technique are needed.

Notice that all the processes only require efficient algorithms (FFT 2D) and standard MIMO methods on a highly reduced subspace.

## 6. Detailed Description of the FFT Compression Technique

In this section, we detail the implementation of the code specifically developed to illustrate the FFT-based approach and its relationship with the directions of various rays (e.g., direct, reflected).

We begin with the original channel matrix, as described in (Equation 2) and illustrated in Figure 2.

Subsequent to applying the FFT transformation, as formulated in (Equation 5), we obtain the matrix $\tilde{H}$, whose sparsified structure is depicted in Figure 3.

To identify the most representative rows and columns of $\tilde{H}$, we define the following summation metrics:(10)$$\begin{array}{ccc} r_{j} & =\sum_{i=1}^{N}\left|\tilde{h}ij\right|,\;c_{i} & =\sum {j=1}^{N}\left|{\tilde{h}}_{ij}\right|, \end{array}$$
where $r_{j}$ and $c_{i}$, respectively, quantify the aggregate magnitude across columns and rows.

Alternative formulations based on the $\ell_{2}$ norm are also considered:(11)$$\begin{array}{ccc} r_{j} & =\sqrt{\sum_{i=1}^{M}{\left|\tilde{h}ij\right|}^{2}},\;c_{i} & =\sqrt{\sum {j=1}^{N}{\left|{\tilde{h}}_{ij}\right|}^{2}}, \end{array}$$
where the differences concerning the $\ell_{1}$-based definition are negligible in practice.

Further, we introduce the maximum row and column metrics:(12)$$\begin{array}{ccc} r_{\max} & =\max_{i=1,\ldots ,N}r_{j},\;c_{\max} & =\max_{i=1,\ldots ,M}c_{i}, \end{array}$$
which serve as benchmarks for the selection of dominant components.

The sets of the most significant rows and columns, $I_{r}$ and $I_{c}$, are then determined according to the thresholding criterion:(13)$$\begin{array}{ccc} I_{r} & =j\in 1,\ldots ,N;|;r_{j}>r_{\max}\delta ,\;I_{c} & =i\in 1,\ldots ,M;|;c_{i}>c_{\max}\delta , \end{array}$$
where δ∈[0.05, 0.1] denotes a tunable sparsification parameter.

Alternatively, a sorting-based approach may be employed. Specifically, the quantities $c_{i}$ and $r_{j}$ are sorted in descending order such that $c_{i(1)}\;\geq \;c_{i(2)}\;\geq \;\ldots \;\geq \;c_{i(N)}$ and $r_{j(1)}\;\geq \;r_{j(2)}\;\geq \;\ldots \;\geq \;r_{j(M)}$, with $i^{\prime }\to i$ and $j^{\prime }\to j$ denoting the corresponding permutation mappings. The representative indices are selected to satisfy(14)$$\begin{array}{ccc} \sum_{i^{\prime }=1}^{n^{\prime }}c_{i}\left(i^{\prime }\right) & >\left(1-\delta \right)\sum_{i=1}^{N}c_{i},\;\sum_{j^{\prime }=1}^{m^{\prime }}r_{j}\left(j^{\prime }\right) & >\left(1-\delta \right)\sum_{j=1}^{M}r_{j}, \end{array}$$
where $n^{\prime }$ and $m^{\prime }$ denote the numbers of selected columns and rows, respectively.

Consequently, the index sets are defined as(15)$$\begin{array}{ccc} I_{r} & =j\left(1\right),j\left(2\right),\ldots ,j\left(m^{\prime }\right),\;I_{c} & =i\left(1\right),i\left(2\right),\ldots ,i\left(n^{\prime }\right). \end{array}$$

Alternatively, a quadratic accumulation criterion may be utilized:(16)$$\begin{array}{ccc} \sum_{i^{\prime }=1}^{n^{\prime }}c_{i}^{2}\left(i^{\prime }\right) & >\left(1-\delta \right)\sum_{i=1}^{N}c_{i}^{2},\;\sum_{j^{\prime }=1}^{m^{\prime }}r_{j}^{2}\left(j^{\prime }\right) & >\left(1-\delta \right)\sum_{j=1}^{M}r_{j}^{2}, \end{array}$$
which emphasizes stronger contributions with higher magnitude.

Using these criteria, we automatically guarantee that at least a (1−δ) fraction of the total angular energy is retained. In other words, $n^{\prime }$ and $m^{\prime }$ grow or shrink with the instantaneous sparsity of $\tilde{H}$, ensuring a balanced trade-off between the compression ratio and estimation accuracy even in rapidly varying NLoS industrial channels [15]. No further per-scenario threshold tuning is required.

Ultimately, the compressed matrix ${\tilde{H}}_{comp}$ is defined as the submatrix of $\tilde{H}$ restricted to the selected rows and columns:(17)$${\tilde{H}}_{comp}=\tilde{H}\left(I_{r},I_{c}\right).$$

**Remark** **3****.**
*The energy-capture criterion in (Equation 14)–(Equation 16) automatically adjusts the number of selected angular components to capture a fixed fraction (1−δ) of the total FFT-domain energy. This self-tuning mechanism ensures robustness to varying channel sparsity, retaining more beams in richly scattered NLoS conditions and fewer in near-LoS cases, without any additional iterative or heuristic threshold tuning.*


A visual representation of the resulting compressed matrix ${\tilde{H}}_{comp}$ is provided in Figure 4.

In this context, we establish the concept of matrix decompression, which involves selectively duplicating the pertinent entries of the matrix (those surpassing the threshold) while setting the remaining matrix elements to zero. We do not decompress the matrix ${\tilde{H}}_{comp}$ to obtain ${\tilde{H}}_{decomp}$; instead, we decompress the SVD decomposition of ${\tilde{H}}_{comp}$, and we use this to approximate the SVD decomposition of $\tilde{H}$:(18)$$\begin{array}{cc} {\tilde{U}}_{comp} & \to {\tilde{U}}_{decomp},\\ {\tilde{V}}_{comp} & \to {\tilde{V}}_{decomp}, \end{array}$$
where(19)$$\begin{array}{cc} \tilde{U}{\left(I_{r},:\right)}_{decomp} & ={\tilde{U}}_{comp}\left(:,:\right),\\ \tilde{V}{\left(I_{c},:\right)}_{decomp} & =\;{\tilde{V}}_{comp}\left(:,:\right). \end{array}$$

Finally, we obtain a fast scheme to obtain the SVD of $\tilde{H}$:(20)$$\tilde{H}\approx {\tilde{H}}_{decomp}=\;{\tilde{U}}_{decomp}{\tilde{\Sigma }}_{comp}{\tilde{V}}_{decomp}^{H}.$$

Notice that ${\tilde{\Sigma }}_{comp}$ does not need to be decompressed, as the rank of $\tilde{H}$ and ${\tilde{H}}_{comp}$ is the same:(21)$${\tilde{\Sigma }}_{decomp}\approx {\tilde{\Sigma }}_{comp}.$$

Now, we go back to the expression $\tilde{H}=F_{2}^{-1}HF_{2}$ and substitute the previous factorization and obtain(22)$$\begin{array}{cc} F_{2}^{-1}HF_{2} & =\tilde{H},\\ F_{2}^{-1}HF_{2} & \approx {\tilde{H}}_{decomp},\\ F_{2}^{-1}HF_{2} & \approx {\tilde{U}}_{decomp}{\tilde{\Sigma }}_{comp}{\tilde{V}}_{decomp}^{H},\\ H & \approx F_{2}{\tilde{U}}_{decomp}{\tilde{\Sigma }}_{comp}{\tilde{V}}_{decomp}^{H}F_{2}^{-1},\\ H & \approx \left(F_{2}{\tilde{U}}_{decomp}\right){\tilde{\Sigma }}_{comp}{\left(F_{2}{\tilde{V}}_{decomp}\right)}^{H}. \end{array}$$

Therefore, we obtain an approximated SVD decomposition of the original channel matrix:(23)$$\begin{array}{cc} H & =U\Sigma V^{H}\approx \left(F_{2}{\tilde{U}}_{decomp}\right){\tilde{\Sigma }}_{comp}{\left(F_{2}{\tilde{V}}_{decomp}\right)}^{H}. \end{array}$$

Therefore,(24)$$\begin{array}{cc} U & \approx F_{2}{\tilde{U}}_{decomp},\\ V & \approx F_{2}{\tilde{V}}_{decomp},\\ \Sigma & \approx {\tilde{\Sigma }}_{decomp}. \end{array}$$

This approach has been implemented to illustrate the relationship between the FFT phase fronts and the directions of various rays (direct, reflected, etc.). We observe a sparsification of the matrix $\tilde{H}$ and a compression of the matrix ${\tilde{H}}_{comp}$, as shown in Figure 2, Figure 3 and Figure 4.

In summary, this section has provided a detailed explanation of how the channel matrix H can be compressed in the context of mMIMO for complex environments using the FFT. This method proves highly beneficial for fast and efficient channel estimation, rapid channel compression, and efficient data transmission. As we demonstrate in the following sections, this technique enables us to achieve transmission bit rates close to Shannon’s limit, while maintaining low complexity in terms of computational requirements, memory usage, and the number of pilots needed for channel estimation.

### 6.1. Block Diagram Description of the Process

The compression process described previously is now described using simple block diagrams. Figure 5 summarizes how the compression technique works.

The figure illustrates a feasible communication method, demonstrating that the channel matrix H can be efficiently compressed into a small number of eigenchannels by employing the FFT and the threshold-based compression technique previously described in detail.

Each eigenchannel corresponds to a specific phase front within the antenna arrays at both the transmitter and receiver and an individual propagation path in the ray-tracing model of the channel matrix H.

More precisely, Figure 6 depicts the channel matrix H compression process in the spatial domain, mapping the transmit antenna elements to the receive antenna elements. Subsequently, an FFT block (or an inverse FFT block at the receiver side) is applied, enabling representation of the channel in the spectral (or angular) domain. The corresponding spectral-domain matrix, denoted as $\tilde{H}$, exhibits significant sparsity, as illustrated in Figure 3. Following this transformation, the most relevant columns at the transmitter and the most relevant rows at the receiver are selected, significantly reducing the matrix dimensions and the computational complexity. This process yields a highly compressed matrix, ${\tilde{H}}_{\mathrm{comp}}$, in the compressed domain. Subsequently, conventional MIMO techniques, such as SVD, can decompose the channel into eigenchannels, facilitating optimal transmission strategies based on individual channel gains and noise levels.

A key observation can be made: the initial three processing blocks at the transmitter and the final three blocks at the receiver consist of linear operations. As a result, it is feasible to precompute an equivalent transmission block and an equivalent reception block. This precomputation enables the direct execution of matrix–vector multiplication, eliminating the need for repeated FFT operations and substantially accelerating the transmission process. Consequently, although the FFT plays a critical role during the channel estimation and compression stages, precomputing the complete precoding and decoding blocks facilitates a significantly faster and more efficient communication scheme once these processes are completed.

### 6.2. Full Description of the Protocol

Figure 7 illustrates the proposed FFT-based communication protocol designed for the complete characterization of the T-MIMO channel within a single frequency band. The protocol is divided into three key stages:Angular Component Estimation: Both the BS and the UE transmit non-directional wave pilots. These signals are processed using a 2D FFT followed by truncation, allowing each side to extract the dominant angular components of the channel. This step reduces the problem’s dimensionality and identifies the most significant propagation paths.Channel Characterization: Leveraging the extracted angular information, a focused measurement is performed using directional beams. This results in the estimation of the reduced-dimension channel matrix, denoted as ${\tilde{H}}_{\mathrm{comp}}$.Data Transmission and Reception: With the channel now characterized in the dominant angular domain, data transmission and reception are conducted using the selected angular components for efficient spatial multiplexing.

While the FFT-based approach offers computational benefits, its resolution and accuracy are constrained by processing capabilities, particularly on the CPU.

## 7. Capacity Obtained in Realistic Scenarios Using FFT Compression

Numerical results on the channel capacity achieved using the FFT-based compression technique in realistic indoor and outdoor scenarios are presented. These include both LoS and NLoS conditions, operating at different frequencies (12 GHz, 24 GHz, and 30 GHz), and varying numbers of antennas at the transmitter and receiver sides.

Figure 8 shows that the FFT-based algorithm performs well in indoor environments. The results highlight that operating with larger arrays even under high compression is more effective than with smaller arrays and lower compression. Larger antenna configurations consistently lead to higher capacity.

Figure 9 demonstrates the performance of FFT-based compression in outdoor scenarios at 12 GHz and 30 GHz, respectively. These results are even more favorable than those indoors, which is expected due to the smaller number of significant propagation paths in outdoor environments.

Figure 10 shows that the FFT-based method also performs well in outdoor scenarios with asymmetric antenna arrays. Again, the results improve with higher frequency, reinforcing the benefit of using FFT compression in millimeter-wave bands, where the ray-tracing approximation is more accurate.

To summarize, the FFT-based channel compression method consistently yields high channel capacities across various environments and configurations. The method benefits from the sparsity of the angular domain representation and the efficiency of the FFT algorithm, making it a practical and scalable solution for mMIMO systems. Performance is powerful in outdoor and high-frequency scenarios, making it a compelling choice for 5G and beyond.

## 8. Computational Complexity of 2D FFT and Viability in Dynamic Urban Traffic

We evaluated the CPU time for a single n × n 2D FFT on a standard laptop. Figure 11 presents these measurements (in microseconds) for array sizes n=4,8,…,32. The total number of antennas in the transmitter is $N_{t}=n\;\times \;n$.

These timings correspond to processing a single OFDM subcarrier. In practice, the dominant angular directions vary slowly across frequency, so support identification needs only to be performed on a small set of widely spaced subcarriers rather than on every subcarrier.

In urban-traffic mmWave channels, coherence times typically range from 20 to 80 ms, i.e., dozens of OFDM frames, before the dominant propagation paths change significantly [16]. Even in the worst case (28 × 28, about 4.7 μs), the FFT occupies only a small fraction of that interval. Thus, we have

Minimal per-frame overhead. A 0.5–5 μs FFT on a single subcarrier can be amortized over M=20–80 frames (20–80 ms), easily fitting within standard subframe budgets.Pilot reduction. One omni-directional pilot plus FFT per estimation round replaces *n* beam-sweeping pilots, yielding an *n*-fold reduction in pilot overhead.**Frequency sampling.** Because angular support is stable across the band, FFT-based support updates need only be run on a few widely separated subcarriers, further reducing the total computational burden.Real-time viability. With only 3–5 dominant paths persisting for tens of milliseconds, periodic FFT support updates incur negligible CPU load, making the method well suited for real-time vehicular deployments.

These results confirm that the computational cost of 2D FFTs is very modest, even in fast-changing urban-traffic channels, and does not impede the practical deployment of our FFT-based angular-domain compression scheme.

### Complexity Analysis and Comparison with Prior Schemes

Table 1 provides both theoretical and empirical per-frame complexity for our FFT-based compression and several representative prior methods.

Empirical timing on a standard laptop confirms that our FFT-only pipeline completes in 0.5–5 μs per subcarrier (see Section 8), which is orders of magnitude faster than a single full SVD and significantly lower than iterative CS or deep-network methods. This lightweight complexity profile makes our method ideal for real-time resource-constrained UE deployments.

## 9. Conclusions

This work has presented an overview of the key challenges and proposed solutions associated with the T-MIMO problem in future wireless communication systems. As T-MIMO is expected to play a pivotal role in 6G networks, leveraging large antenna arrays at the base station to enhance spectral efficiency significantly, the complexity associated with acquiring and feeding back CSI emerges as a critical bottleneck, particularly in the 10–30 GHz frequency range and mmW bands.

We introduced and validated a novel compression technique based on the FFT to address this issue. This approach projects the channel matrix H into the angular domain, where it exhibits pronounced sparsity. By exploiting this sparsity, the method selectively retains only the most significant angular components, corresponding to dominant physical propagation paths between the transmitter and receiver, thereby substantially reducing the problem’s dimensionality.

We demonstrated that the FFT-based transformation yields an equivalent channel matrix that facilitates simplified precoding and detection, significantly lowering the Uplink (UL) feedback overhead while maintaining high CSI quality. The proposed method is particularly well-suited for high-frequency operation, where ray-tracing approximations hold and dominant propagation paths are more easily distinguishable.

Numerical evaluations conducted in both indoor and outdoor environments confirm that the FFT-based compression technique achieves near-optimal channel capacity even under significant compression. The method proves especially effective in outdoor scenarios and at higher frequencies, where fewer dominant rays and improved angular resolution are observed.

In conclusion, FFT-based channel compression emerges as a robust and scalable strategy for reducing the complexity of mMIMO systems. It enables efficient transmission, reduces feedback requirements, and aligns well with realistic deployment scenarios anticipated for 5G and future wireless generations.

## Figures and Tables

**Figure 1 sensors-25-04544-f001:**
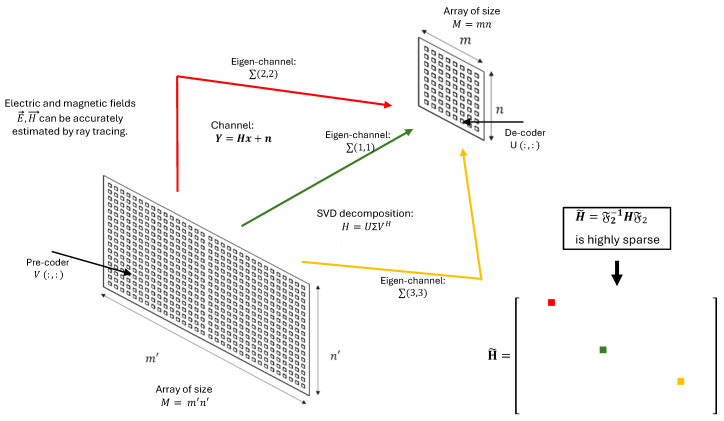
Depiction of various physical routes connecting the transmitter and receiver.

**Figure 2 sensors-25-04544-f002:**
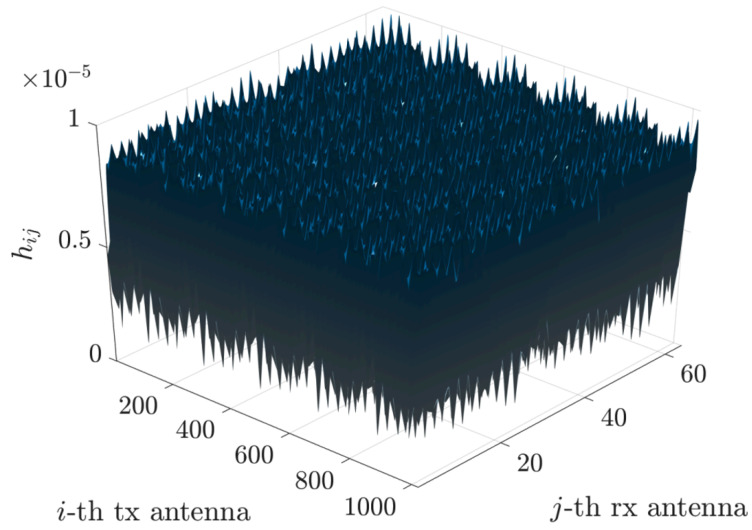
Full channel matrix.

**Figure 3 sensors-25-04544-f003:**
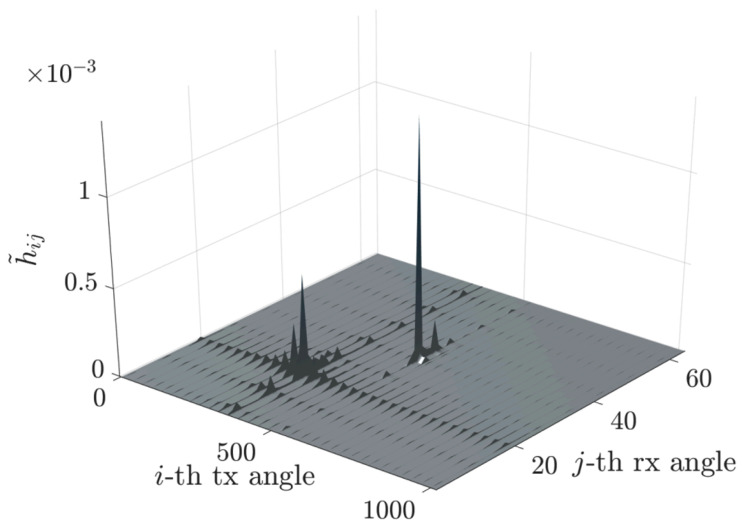
Sparsified channel matrix $\tilde{H}$.

**Figure 4 sensors-25-04544-f004:**
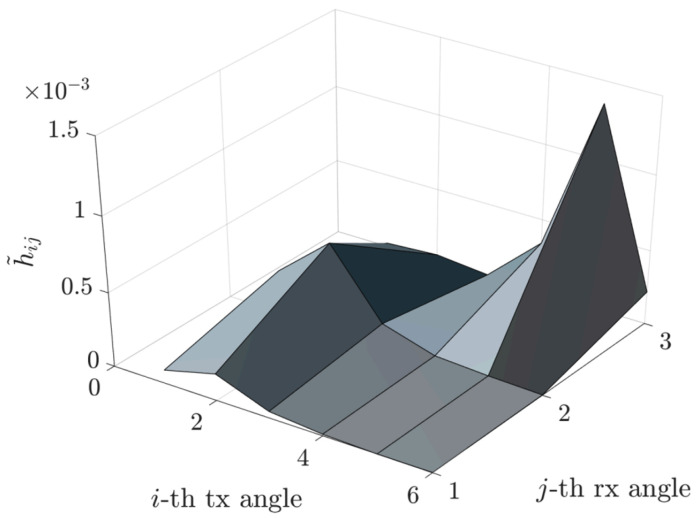
Sparse compressed channel matrix ${\tilde{H}}_{comp}$.

**Figure 5 sensors-25-04544-f005:**
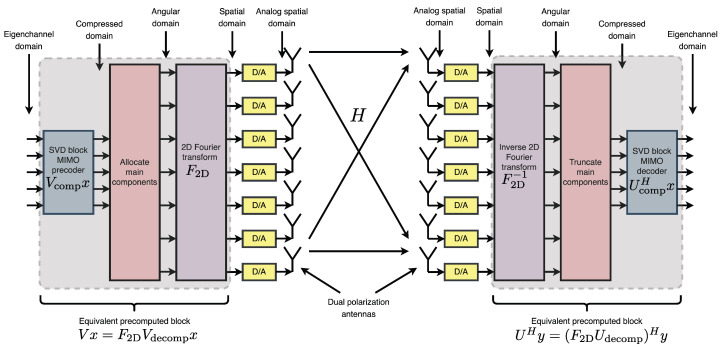
Full description of the compressibility of the channel matrix via FFT.

**Figure 6 sensors-25-04544-f006:**
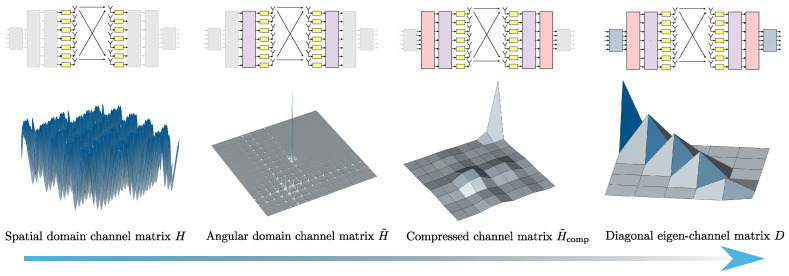
Matrix H compression process.

**Figure 7 sensors-25-04544-f007:**
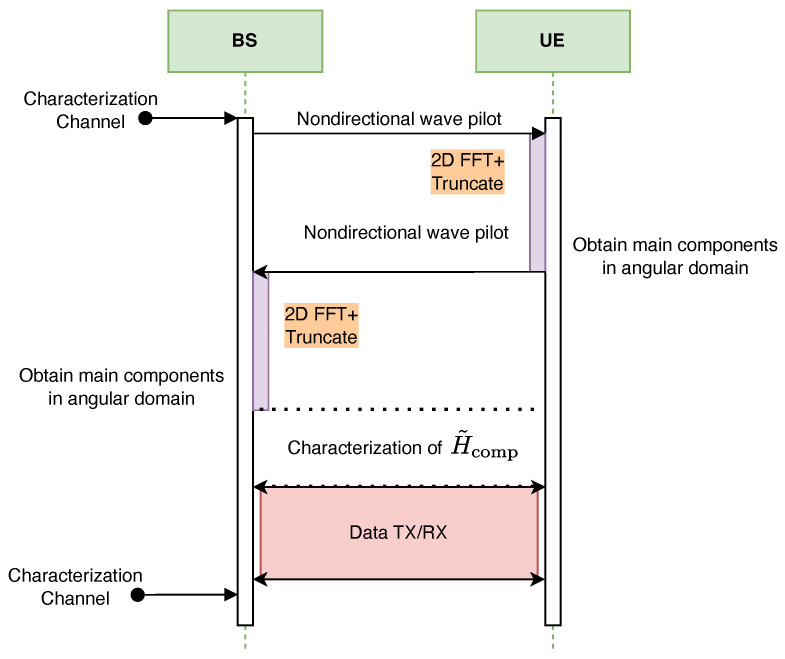
Full cycle of T-MIMO channel characterization using FFT, followed by data transmission.

**Figure 8 sensors-25-04544-f008:**
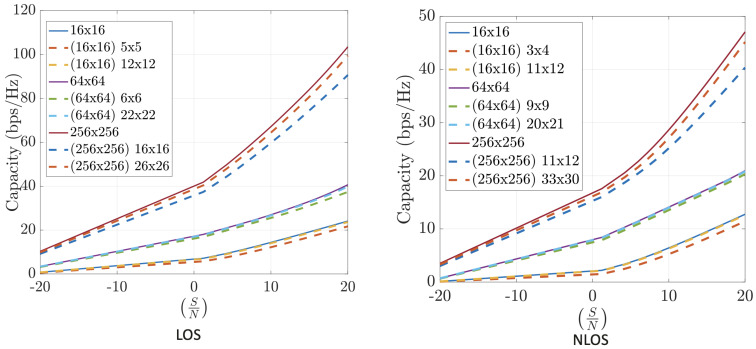
Capacity estimation for FFT-compressed channel at 12 GHz. INDOOR. Line-of-sight (**left**) and non-line-of-sight (**right**).

**Figure 9 sensors-25-04544-f009:**
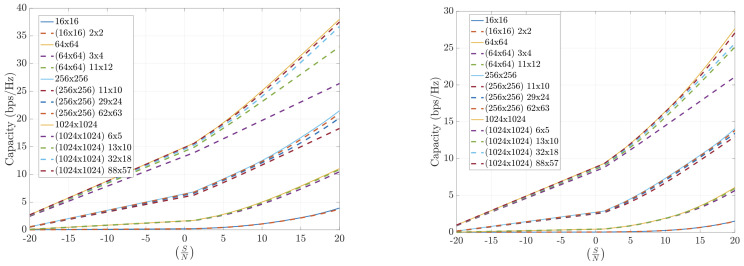
Capacity estimation for FFT-compressed channel in outdoor scenarios at 12 GHz (**left**) and 30 GHz (**right**). Symmetric arrays (N=M).

**Figure 10 sensors-25-04544-f010:**
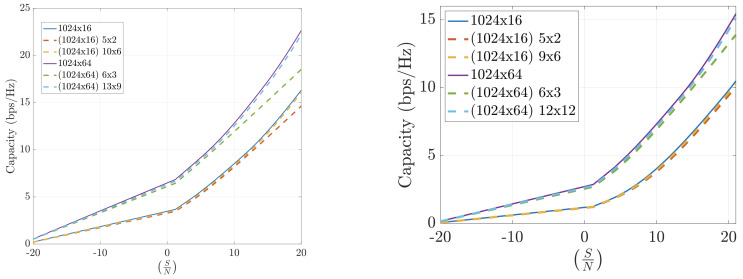
Capacity estimation for FFT-compressed channel in outdoor scenarios at 12 GHz (**left**) and 30 GHz (**right**). Asymmetric arrays (N≠M).

**Figure 11 sensors-25-04544-f011:**
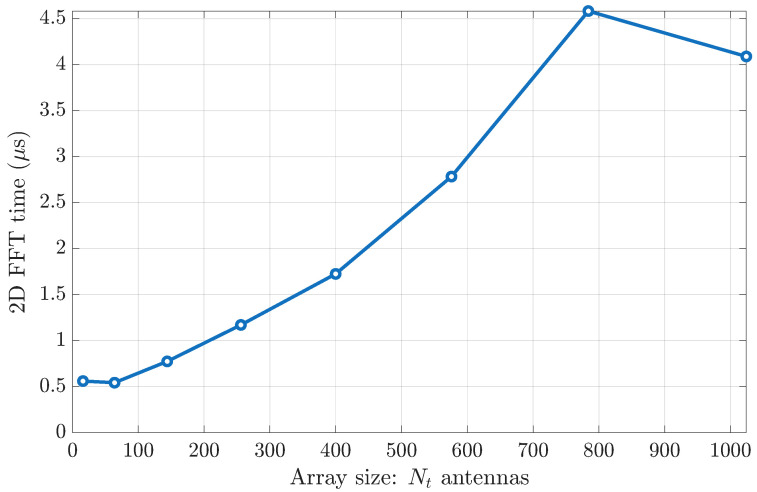
CPU time per 2D FFT vs. array size n × n.

**Table 1 sensors-25-04544-t001:** Complexity comparison of CSI compression schemes (per OFDM subcarrier).

Method	Computational Complexity	Remarks
FFT-only (this work)	O(NlogN)	0.5–5 μs per subcarrier; no iterations or training
Full-matrix SVD (no compression)	$O(N^{3})$	Impractical for large *N*
Iterative CS ([1])	O(TNlogN)	*T* FFTs + matrix ops per iteration
Unrolled network (LORA, [2])	$O(D_{\inf})$	On-device inference + codebook lookups
Diffusion-based ([3])	$O(D_{\inf}\;\times \;K)$	*K* denoising steps per reconstruction

## Data Availability

The results depicted in the paper are theoretically calculated and can be reproduced with the formulation included in the paper.

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
