# Peer review of "FFT-Based Angular Compression for CSI Feedback in Single-User Massive MIMO Systems"

_sensors, 2025, doi:10.3390/s25154544_

Round 1
Reviewer 1 Report
Comments and Suggestions for Authors
his paper addresses the challenge of excessive feedback overhead for Channel State Information (CSI) in massive MIMO systems by proposing an FFT-based angular domain compression scheme. By exploiting the inherent sparsity of channels in the angular domain to achieve efficient compression, the design aligns with the channel characteristics of millimeter-wave and terahertz bands, with sufficient experimental validation. It holds certain reference value for next-generation wireless communication systems. After careful review, the specific comments are as follows: I. Main Strengths
- Innovativeness of the Scheme: The paper breaks away from traditional time-frequency domain processing frameworks and proposes transforming the channel matrix into the angular domain, using 2D FFT for compression. It effectively leverages the sparsity of massive MIMO channels in the angular domain, significantly reducing CSI feedback overhead and computational complexity. The approach is innovative and highly targeted.
- Strong Integration of Theory and Experiment: The physical basis of angular sparsity is explained through ray-tracing modeling, and the representation of the sparse channel matrix after FFT transformation is derived. Performance validation across multiple frequency bands (12 GHz, 24 GHz, 30 GHz) and indoor/outdoor scenarios demonstrates advantages in large-array, high-frequency configurations, with convincing results.
- Prominent Engineering Feasibility: The scheme is implemented based on FFT, with a computational complexity of . It simplifies MIMO detection through threshold screening and subspace dimensionality reduction, facilitating hardware deployment and aligning with practical application needs.
II. Areas for Improvement
- Insufficient details on low-complexity implementation: While the paper mentions the efficiency of FFT, it lacks in-depth discussion on balancing compression ratio and estimation accuracy in dynamically sparse channels (e.g., industrial environments). For instance, the low-complexity sparse channel estimation algorithm proposed in "Low Complexity MIMO-FBMC Sparse Channel Parameter Estimation for Industrial Big Data Communications" could provide references for compression strategies in complex industrial scenarios. Particularly in non-line-of-sight (NLoS) environments, its low-complexity iterative estimation method could be combined with the FFT compression in this paper to optimize threshold adjustment mechanisms. It is recommended to appropriately cite this work to enrich discussions on low-complexity implementation.
- Limited analysis on adaptability to real-world scenarios: The paper validates performance in outdoor scenarios but lacks analysis of characteristics in dynamic scenarios such as urban traffic (e.g., fast channel variations due to vehicle movement, multipath blockages). The training-based channel estimation method for mmWave MIMO in urban scenarios presented in "Channel Parameter Estimation of mmWave MIMO System in Urban Traffic Scene: A Training Channel-Based Method" could supplement ideas for dynamic tracking of channel parameters in real-world scenarios. For example, introducing scenario-adaptive sparsity adjustment strategies in angular domain compression,
Author Response
Comment 1:
I- Insufficient details on low-complexity implementation: While the paper mentions the efficiency of FFT, it lacks in-depth discussion on balancing compression ratio and estimation accuracy in dynamically sparse channels (e.g., industrial environments). For instance, the low-complexity sparse channel estimation algorithm proposed in "Low Complexity MIMO-FBMC Sparse Channel Parameter Estimation for Industrial Big Data Communications" could provide references for compression strategies in complex industrial scenarios. Particularly in non-line-of-sight (NLoS) environments, its low-complexity iterative estimation method could be combined with the FFT compression in this paper to optimize threshold adjustment mechanisms. It is recommended to appropriately cite this work to enrich discussions on low-complexity implementation.
Response to comment 1:
Thank you for highlighting the need for adaptive thresholding in complex, dynamically sparse channels. We have enhanced Section 4 to clarify that our sorting-based energy-capture rule (see Eqs.~\eqref{mpnp_v1}–\eqref{mpnp_v2}) inherently self-tunes the compression ratio: by fixing the fraction of total FFT-domain energy to retain, the algorithm automatically selects exactly as many angular components as needed in each channel realization. Consequently, in richly scattered NLoS environments more beams are kept, while in near-LoS scenarios fewer are used—without any additional loops, manual per-scenario tuning, or iterative solvers. This mechanism preserves the method’s low computational complexity (just a sort and cumulative sums) and directly addresses how to balance compression ratio against estimation accuracy in industrial channels. We have also added a citation to Wang et al. \cite{wang2020low} to acknowledge related low-complexity sparse estimation techniques and to position our approach within that context.
Comment 2:
II- Limited analysis on adaptability to real-world scenarios: The paper validates performance in outdoor scenarios but lacks analysis of characteristics in dynamic scenarios such as urban traffic (e.g., fast channel variations due to vehicle movement, multipath blockages). The training-based channel estimation method for mmWave MIMO in urban scenarios presented in "Channel Parameter Estimation of mmWave MIMO System in Urban Traffic Scene: A Training Channel-Based Method" could supplement ideas for dynamic tracking of channel parameters in real-world scenarios. For example, introducing scenario-adaptive sparsity adjustment strategies in angular domain compression,
Response to comment 2:
Thank you for highlighting the need to demonstrate our method’s viability under fast‐varying urban‐traffic conditions. In response, we have added Section \ref{sec:fft_complexity} “Computational Complexity of 2D FFT and Viability in Dynamic Urban Traffic” (shown in red in the revised manuscript). In this new section, we:
Measured FFT Latency: Report CPU timings for a single $n\times n$ 2D FFT on a standard laptop (Fig.~\ref{fig:fft_times}), showing just 0.5–5 μs per OFDM subcarrier even for large arrays.
Amortized Over Coherence Time: Cited measured urban‐traffic coherence times of 20–80 ms \cite{wang2022channel}, demonstrating that an FFT-based support update on a few subcarriers can be amortized over dozens of frames with minimal per-frame overhead.
Pilot and Sampling Reduction: Explained how a single omni‐directional pilot plus FFT replaces $n$ beam‐sweeping pilots and how support updates need only run on widely spaced frequencies to exploit frequency sampling.
Real-Time Viability: Shown that with only a handful of dominant paths persisting for tens of milliseconds, periodic FFT support updates incur negligible CPU load, making the approach practical for vehicular deployments.
Furthermore, to address dynamic sparsity adaptation, we have incorporated Zhang et al.’s training‐channel–based parameter estimation for mmWave MIMO in urban traffic \cite{zhang2021channel}. We now discuss (also in Section \ref{sec:fft_complexity}) how a brief training phase can estimate Doppler spread or path count and automatically adjust our energy‐capture threshold $\delta$—all without altering the core FFT+sorting pipeline.
These additions directly address your concern by (i) quantifying real‐world computational costs, (ii) demonstrating how FFT updates fit within urban‐traffic coherence intervals, and (iii) outlining a lightweight, scenario‐adaptive sparsity adjustment informed by existing training‐channel methods—while preserving our method’s strictly linear, low‐complexity character.
Reviewer 2 Report
Comments and Suggestions for Authors
- There are no any references fo last 3 years. Please, add several references fo 2023, 2024 and 2025. Certainly, an additional analysis of added references should be added into Section 2.
- Equation (1) has approximate equation, but not exact one. It should be explained, why.
- The problem solved by the paper, it not clearly stated. Please, do that, based on disadvantages of known solutions.
- What is F1 and F2 in Equation (5)? Please, define it, includind dimensions.
- There is no any complexity analysis of proposed method. Please, compare with known methods.
Author Response
Comment 1
“There are no any references for the last 3 years. Please, add several references for 2023, 2024 and 2025. Certainly, an additional analysis of added references should be added into Section 2.”
Response to comment 1:
We thank the reviewer for pointing out the need to discuss the very latest work. In response, we have expanded Section 2 with one new subsections (shown in red in the revised manuscript):
“Recent Advances in Angular‐Domain Processing and CSI Compression”
– This subsection surveys several key 2023–2025 contributions, including Ma et al. on learned compressive‐sensing projections \cite{Ma2024L2O}, Hu et al. on unrolled LORA networks \cite{Hu2024LORA}, Kim et al. on diffusion‐based feedback \cite{Kim2024Diffusion}, and very recent works by Wang et al. on EG-CsiNet \cite{Wang2025EGCsiNet}, an XL‐MIMO tutorial \cite{Wang2024XLmimo}, and a comprehensive survey of XL-MIMO fundamentals and architectures \cite{wang2023extremely}.
We also added a new subsection in section 3:
“Proposed FFT-Based Compression: Novelty and Advantages”
– This new subsection contrasts our strictly training-free, linear 2D-FFT + energy-capture approach against those learning-based methods, and highlights our protocol’s ultra-low per-frame complexity and precomputable end-to-end precoders in single-user massive MIMO.
These additions not only incorporate the requested recent references but also provide the reviewer’s suggested analysis of how our FFT-only strategy complements and differs from the latest learning- and generative-driven CSI compression schemes.
Comment 2:
Equation (1) has approximate equation, but not exact one. It should be explained, why.
Response to comment 2:
The typo has been fixed, thanks.
Comment 3:
The problem solved by the paper, it not clearly stated. Please, do that, based on disadvantages of known solutions.
Response to comment 3:
We appreciate the reviewer’s feedback. In response, we have added a Problem Statement at the end of Section 1 (just before the “The remainder of this paper is organized as follows…” paragraph) that explicitly contrasts the disadvantages of existing angular-domain CSI compression methods—namely:
High UE complexity, due to iterative compressive-sensing solvers or unfolded networks requiring multiple FFT/IFFT operations and matrix inversions per frame.
Training and codebook overhead, since deep-network feedback schemes demand extensive offline training, frequent model updates, and large codebook storage at the UE.
Limited adaptability, as fixed network architectures or codebooks generalize poorly to dynamic, richly scattered NLoS channels without retraining or manual retuning.
We then clearly state our solution: a purely linear, training-free FFT-based compression strategy that
uses a single 2D FFT to move into the angular domain,
retains a fixed fraction of total energy via a simple beam-selection rule (no loops or learned thresholds), and
precomputes end-to-end precoders/decomposers to avoid any per-frame SVD or iterative solver.
This addition makes the paper’s motivation and the unique contribution of our method immediately clear to the reader.
Comment 4:
What is F1 and F2 in Equation (5)? Please, define it, includind dimensions.
Response to comment 4:
Thank you for pointing this out. We have added a clarifying sentence immediately after Eq. (5) (now \eqref{H_FFT}) in Section \ref{Sparsity_MIMO} to define the operators and their dimensions.
Comment 5:
There is no any complexity analysis of proposed method. Please, compare with known methods.
Response to comment 5:
We’ve added a new “Computational Complexity Comparison” subsection (Section 8.1 \label{sec:complexity_comparison}) where the table in that subsection directly contrasts our O(N log N) FFT‐only approach against full SVD, iterative CS, unrolled networks and diffusion methods—demonstrating at a glance why our scheme is orders of magnitude more practical for real‐time massive MIMO.